# Lactate dehydrogenase can be used for differential diagnosis to identify patients with severe polytrauma with or without chest injury—A retrospective study

**Weining Yan**[1,2][�]*, **Felix Bläsius**[1☺], **Tabea Wahl**[2], **Frank Hildebrand**[1], **Elizabeth Rosado Balmayor**[2], **Johannes Greven**[2‡]*, **Klemens Horst**[1‡]

**1** Department of Orthopaedics, Trauma and Reconstructive Surgery, University Hospital RWTH Aachen, Aachen, Germany, **2** Experimental Orthopaedics and Trauma Surgery, Department of Orthopaedics, Trauma and Reconstructive Surgery, University Hospital RWTH Aachen, Aachen, Germany

☺ These authors contributed equally to this work.
‡ JG and KH also contributed equally to this work.
* wyan@ukaachen.de (WY); jgreven@ukaachen.de (JG)

**Data Availability Statement:** All relevant data are within the manuscript and its Supporting

## Abstract

### Background

Chest injury is an important factor regarding the prognosis of patients with polytrauma (PT), and the rapid diagnosis of chest injury is of utmost importance. Therefore, the current study focused on patients' physiology and laboratory findings to quickly identify PT patients with chest injury.

### Method

Data on 64 PT patients treated at a trauma center level I between June 2020 and August 2021 were retrospectively collected. The patients were divided into a PT group without chest injury (Group A) and a PT group including chest injury (Group B). The relationship between chest injury and the patients' baseline characteristics and biochemical markers was analyzed.

### Results

Heart rate, respiration rate, Sequential Organ Failure Assessment (SOFA) score, glutamate oxaloacetate aminotransferase (GOT), glutamate pyruvate transaminase (GPT), creatine kinase MB (CK-MB), leucocytes, hemoglobin (Hb), platelets, urine output, lactate, and lactate dehydrogenase (LDH) in groups A and B exhibited statistically significant differences at certain time points. Multifactorial analysis showed that blood LDH levels at admission were associated with chest injury (P = 0.039, CI 95% 1.001, 1.022).

### Conclusion

LDH may be a promising indicator for screening for the presence of chest injury in patients with severe polytrauma.

information files. Corresponding author can be contacted for any other requirements.

**Funding:** The author(s) received no specific funding for this work.

**Competing interests:** The authors have declared that no competing interests exist.

## Introduction

Trauma is considered one of the most important public health problems worldwide, accounting for approximately 5.8 million deaths per year, reflecting 10% of the global mortality rate [1]. Thereby patients with polytrauma (PT) represent a challenging population due to the complexity of their injuries [2]. Chest injuries occur commonly in this population and are considered to have a high priority in primary care because of the respiratory or circulatory complications they can cause. Furthermore, aside from severe traumatic brain injury, lethal hemorrhage, and abdominal injuries, chest trauma is a common cause of death in PT patients [3]. In patients with PT, assessing the presence of chest trauma and its severity therefore has a significant influence on the further treatment strategy [4]. The commonly used tools currently available for the assessment of chest trauma are physical examination, computed tomography (CT), and ultrasound [5, 6]. These tests rely heavily on patient feedback and examination facilities, so having a reliable predictive model to quickly and accurately screen patients with severe polytrauma for the presence of chest injuries remains a necessity for current emergency and clinical care. Although, lactate levels can reflect tissue hypoxia and be used as a prognostic indicator in trauma patients [7], lactate's significance for the diagnosis of chest injury is unclear. Lactate dehydrogenase (LDH) however, also can be used as a marker of cellular damage for the diagnosis of severe tissue injury [8, 9]. Yet, due to its lack of tissue specificity, there are no reports of its use in the diagnosis of chest injury.

## Method

### Research population and ethical permission

Data were retrospectively collected on all patients with PT who were treated for severe trauma at the RWTH Aachen University Hospital Trauma Surgery Department (Trauma Level Center I) between June 2020 and August 2021. Data collection was approved by the Ethical Committee of RWTH Aachen University Hospital (no. EK 401–19, the original document has been submitted). Clinical data were collected and anonymized by specialized person (TW, and TW was not involved in the analysis of the data or the interpretation of the results). The inclusion criteria were patients diagnosed with severe PT. The following exclusion criteria were applied: age <18 years, diagnosis of PT not fulfilled (see below), patient discharged from the emergency room, death on arrival at the hospital or within the first 24 h, participation in other studies, radiation or chemotherapy within the last 3 months, immunosuppression, kidney dialysis, reanimation at place of injury, pregnancy, or incomplete medical data. A total of 64 patients were included in the study (Fig 1).

### Definitions

Trauma severity was rated using the abbreviated injury scale (AIS) and the ISS, an internationally accepted scoring system. Severe polytrauma is defined as significant injuries of three or more points in two or more different anatomic AIS regions [10, 11]. Chest injury is defined as a patient with a chest AIS ≥1. The AIS score (0–5) is assigned to 6 body regions: head, face, chest, abdomen, the distal region (including the pelvis), and the body surface. The scores of the 3 most severely injured body regions are squared and summed to obtain an ISS score (0–75). If the AIS for any area is 6 (unsurvivable injury), the ISS is automatically scored as 75 [12].

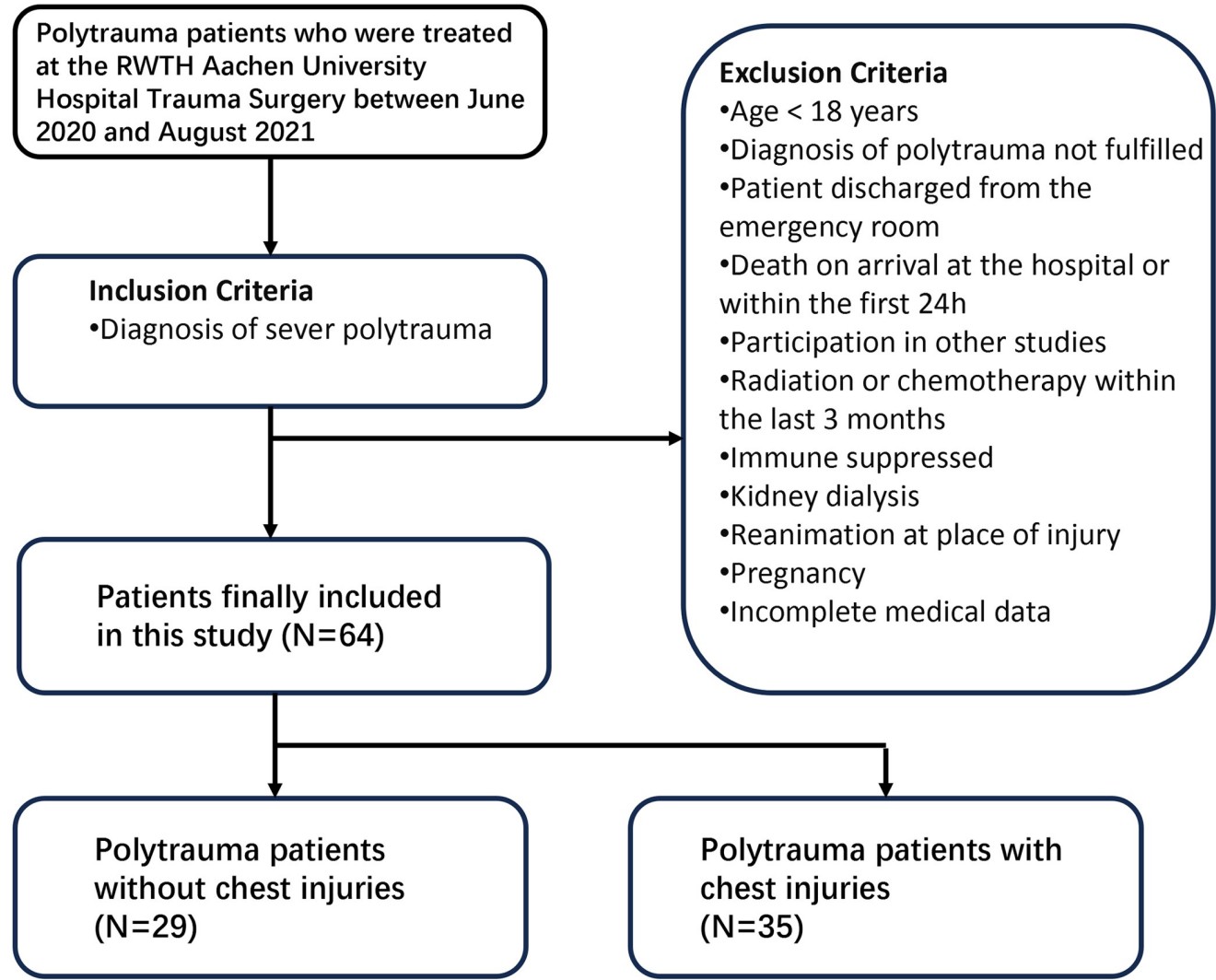

**Fig 1. Inclusion and exclusion criteria for the study population.** This figure shows the source of the study population, the inclusion and exclusion criteria, and the grouping of patients in this study.

## Data collection

The following information was extracted from the medical records and electronic laboratory results of the included patients: heart rate (HR), respiration rate (RR), age, time in hospital, Injury Severity Score (ISS), Glasgow Coma Scale (GCS) score, Sequential Organ Failure Assessment (SOFA) score, presence of organ failure, volume of blood transfused, urine output, hemoglobin (Hb), leucocytes, platelets, hematocrit, C-reactive protein (CRP), potassium ions ($K^+$), calcium ions ($Ca^{2+}$), cholesterol, glutamate oxaloacetate aminotransferase (GOT), glutamate pyruvate transaminase (GPT), creatinine, creatine kinase (CK), creatine kinase MB (CK-MB), pH, lactate, lactate dehydrogenase (LDH), alkaline phosphatase (ALP), carbon dioxide partial pressure ($PCO_2$), oxygen partial pressure/fraction of inspired oxygen ($PO_2/FiO_2$), bicarbonate ($HCO_3^-$), partial prothrombin time (PPT), and international normalized ratio (INR). The assessment scores and biochemical results were available on admission (0 h/shock

**Table 1. Summary of patient information data.**

| Variable | |
|---|---|
| HR* | Ca$^{2+}$* |
| RR* | Cholesterol* |
| age | GOT* |
| time in hospital | GPT* |
| ISS* | Creatinine* |
| GCS* | CK* |
| SOFA* | CK-MB* |
| presence of organ failure | pH* |
| volume of blood transfused | Lactate* |
| urine output* | LDH* |
| Hb* | ALP* |
| Leucocyte* | PCO$_2$* |
| Platelet* | PaO$_2$/FiO$_2$* |
| Hematocrit* | HCO$_3^-$* |
| CRP* | PPT* |
| K$^+$* | INR* |

This table shows all the clinical and laboratory information that was collected on the study population and the point in time at which the information was collected.

* Data collected corresponded to admission (0h/shock room), 8h post admission and, first to tenth day (every 24h) for a total of 12 time points.

HR: Heart rate; RR: Respiration rate; ISS: Injury Severity Score; GCS: Glasgow Coma Scale; SOFA: Sequential Organ Failure Assessment; Hb: Hemoglobin; CRP: C-reactive protein; K$^+$: Potassium ions; Ca$^{2+}$: Calcium ions; GOT: Glutamate oxaloacetate aminotransferase; GPT: Glutamate-pyruvate transaminase; CK: Creatine kinase; CK-MB: Creatine Kinase-MB; LDH: Lactate dehydrogenase; ALP: Alkaline phosphatase; PO$_2$: Oxygen partial pressure; PCO$_2$: Carbon dioxide partial pressure; FiO$_2$: Fraction of inspiration Oxygen; HCO$_3^-$: Bicarbonate; PPT: Partial prothrombin time; INR: International normalized ratio.

room), 8 h post admission, and on the first to the tenth day (every 24 h) for a total of 12 time points (Table 1).

## Statistical approaches and data analysis

The continuous variables that followed normal distribution after the Shapiro-Wilk test were used to calculate p-values using Student's t-test and the T-test. Among the nonnormally distributed continuous variables, the ordinal variables were subjected to the Wilcoxon rank sum test, and Fisher's exact test or Pearson's chi-square test were used for the categorical variables. Significance was set at p<0.05. We used binary logistic regression analysis to assess the correlation of the clinical data with chest trauma injuries. Furthermore, the statistically significant correlated variables in the univariate analysis were included to determine the efficacy of the various assessment scores and biochemical parameters analyzed in this study for predicting chest injury. For the same parameter at different time points, we selected the time point with the best univariate analysis results. We calculated the associations between the variables included in the multivariate analyses, with significance set at p <0.05. Sensitivity and specificity tests were defined using receiver operating characteristic (ROC) curves to test the feasibility of screening variables for prediction. The statistical analyses were performed using IBM SPSS software v. 25.0 (IBM Corp., Armonk, NY, USA).

## Results

A total of 64 patients with severe PT (ISS = 30.71 [range 17–57]) were included in this study, of whom 29 were without chest injury (Group A) and 35 were with chest injury (Group B, mean chest AIS = 3.14 [range 3–4]). Demographic results are depicted in Table 1. After comparing all the data between the groups, heart rate, respiration rate, SOFA score, GOT, GPT, CK-MB, leucocytes, Hb, platelets, urine output, lactate, and LDH exhibited statistically significant differences at certain time points (Table 2). Other indicators including age, time in hospital, and ISS, did not differ significantly between the two groups of patients (S1 Table).

From the obtained data of the 64 patients (Table 1), a logistic regression analysis was performed (Table 3). We first performed correlation analyses on all the indicators to filter the covariates for the logistic regression analysis (S2 Table). The results showed that HR, RR, GCS score, SOFA score, urine output, CRP, $K^+$, GOT, GPT, CK-MB, leucocytes, Hb, platelets, lactate, LDH, and ALP were associated with the presence of chest injuries in the polytrauma patients. We performed univariate binary logistic regression analyses of these factors and selected significant factors as covariates in the multivariate logistic regression analyses (Table 3). For the multifactor binary logistic regression analysis, we used the stepwise forward method for regression analysis. Variables not included in the equation are shown by score (residuals/degrees of freedom). The only variable left in the final equation was LDH (OR = 1.011) (Table 3).

**Table 2. Comparison of biochemical markers between patients with severe polytrauma without chest injury (Group A) and with chest injury (Group B).**

| Variable | A Group | B Group | P value |
|---|---|---|---|
| | n = 29 | n = 35 | |
| HR d6 (beats/min) | 77.86±15.08 | 68.2±10.44 | 0.034* |
| RR d1 (times/min) | 14.43±3.56 | 17.41±4.53 | 0.017* |
| SOFA d1 | 7.05±3.69 | 9.35±3.55 | 0.041* |
| Output Urine d3(ml) | 2357.33±622.2 | 3102.17±1172.67 | 0.030* |
| GOT 8h(U/l) | 73.61±65.04 | 187.91±252.14 | 0.015* |
| GOT d1(U/l) | 76.96±70.85 | 198.15±314.56 | 0.036* |
| GOT d7(U/l) | 52.16±29.52 | 82.62±64.42 | 0.019* |
| GOT d8(U/l) | 48.16±21.33 | 70.35±47.73 | 0.020* |
| GPT d1(U/l) | 37±22.27 | 143.85±289.56 | 0.039* |
| GPT d7(U/l) | 43.64±35.71 | 81.65±87.42 | 0.027* |
| CK-MB d1(U/l) | 90.63±70.82 | 152.95±121.04 | 0.046* |
| Leucocyte d3 (U/nl) | 10.54±3.85 | 8.01±2.88 | 0.010* |
| Hb d5 (g/dl) | 9.75±2.24 | 8.67±1.35 | 0.036* |
| Platelet d3(U/nl) | 182.2±81.78 | 139.26±44.77 | 0.023* |
| Platelet d4(U/nl) | 191.88±65.82 | 151.79±53.69 | 0.012* |
| Lactate d1[mg/dl] | 23.88±15.57 | 36.58±26.03 | 0.034* |
| LDH 0h[U/l] | 358.29±191.29 | 533.06±247.23 | 0.004* |
| LDH d1[U/l] | 308.89±72.03 | 466.17±176.94 | 0.014* |

This table shows all the biomarkers that are significantly different between Group A and Group B.

*: P-value<0.05. Data shown as mean ± standard deviation.

0h: on admission; 8h: 8 hours post admission; dx: x days post admission.

HR: Heart rate; RR: Respiration rate; GOT: Glutamate oxaloacetate aminotransferase; GPT: Glutamate-pyruvate transaminase; CK-MB: Creatine Kinase-MB; Hb: Hemoglobin; LDH: Lactate dehydrogenase.

**Table 3. Logistic regression analysis of factors associated with the presence or absence of chest injury in patients with severe polytrauma.**

| Risk factors | Univariate | | Multivariate | |
|:---:|:---:|:---:|:---:|:---:|
| | OR (95% CI) | p-value | Score/OR (95% CI) | p-value |
| HR d6 (beats/min) | 0.938(0.881,0.999) | **0.047**[*] | 0.926 | 0.336 |
| RR d1 (beats/min) | 1.197(1.023,1.399) | **0.025**[*] | 2.869 | 0.09 |
| GCS d1 | 0.896(0.782,1.028) | 0.117 | | |
| SOFA d1 | 1.2(1.001,1.439) | **0.049**[*] | 0.084 | 0.772 |
| Output Urine d3(ml) | 1.001(1,1.002) | **0.038**[*] | 0.608 | 0.435 |
| CRP d6 (mg/dl) | 1.056(0.986,1.131) | 0.121 | | |
| K$^+$ d10 | 0.611(0.242,1.544) | 0.297 | | |
| GOT 8h(U/l) | 1.008(1.001,1.016) | **0.034**[*] | 0.001 | 0.979 |
| GOT d1(U/l) | 1.007(1,1.015) | **0.046**[*] | | |
| GOT d7(U/l) | 1.014(1,1.027) | **0.047**[*] | | |
| GOT d8(U/l) | 1.017(1,1.035) | **0.047**[*] | | |
| GPT d1(U/l) | 1.024(1.003,1.046) | **0.024**[*] | 0.011 | 0.916 |
| GPT d7(U/l) | 1.015(0.999,1.031) | 0.066 | | |
| CK-MB d1(U/l) | 1.007(1,1.015) | 0.067 | | |
| Leucocyte d3 (U/nl) | 0.79(0.647,0.964) | **0.020**[*] | 1.226 | 0.268 |
| Hb d5 (g/dl) | 0.714(0.525,0.97) | **0.031**[*] | 2.583 | 0.108 |
| Lactate d1[mg/dl] | 1.031(1,1.062) | **0.048**[*] | 1.028 | 0.311 |
| LDH 0h[U/l] | 1.005(1.001,1.008) | **0.011**[*] | 1.011(1.001,1.022) | **0.039**[*] |
| LDH d1[U/l] | 1.010(1.000,1.021) | **0.047**[*] | | |
| ALP d8[U/l] | 1.007(0.989,1.025) | 0.438 | | |

This table presents the outcomes of binary logistic regression analyses, identifying severe polytrauma with concurrent chest injury as the outcome of interest. Initially, a binary logistic univariate analysis was conducted on all variables correlated with the outcome, as determined by correlation tests. Subsequently, variables that demonstrated significance in the univariate analyses were incorporated into a binary logistic multivariate analysis, employing a forward stepwise selection method. The findings highlight LDH as an independent risk factor for the presence of concurrent chest injuries in severe polytrauma patients.

[*]: P-value<0.05.

0h: on admission; 8h: 8 hours post admission; dx: x days post admission.

HR: Heart rate; RR: Respiration rate; ISS: Injury Severity Score; GCS: Glasgow Coma Scale; SOFA: Sequential Organ Failure Assessment; Hb: Hemoglobin; CRP: C-reactive protein; K$^+$: Potassium ions; GOT: Glutamate oxaloacetate aminotransferase; GPT: Glutamate-pyruvate transaminase; CK-MB: Creatine Kinase-MB; LDH: Lactate dehydrogenase; ALP: Alkaline phosphatase.

ROC curves were used to examine the sensitivity, specificity, and threshold values of LDH for screening patients with severe multiple injuries accompanied by chest injury (Fig 2). ROC curves were drawn considering polytrauma with chest injury as a positive result and polytrauma without chest injury as a negative result to obtain the optimal cutoff point of LDH = 365 U/l (p<0.001). Table 4 presents the sensitivity, specificity, positive predictive value, and negative predictive value of LDH as a predictor of the presence of concomitant chest injury in patients with multiple injuries when the LDH threshold is set at 365 U/L.

Subsequently, to identify the source of chest injuries associated with elevated LDH levels, we stratified the patients into two cohorts using an LDH threshold of 365 U/l and examined the differences in the incidence of various chest injuries. Due to the complexity of the diagnoses, we categorized the injuries as follows: fractures of the chest wall (including ribs, sternum, and thoracic vertebrae), extra-thoracic wall fractures (including scapula and clavicle), pulmonary contusion, pleural diseases (including pneumothorax, haemothorax, and pleural effusion), pneumonia, and cardiovascular diseases (including blunt aortic injury, myocardial contusion, pericardial effusion, and mediastinal haematoma). Our results indicated that

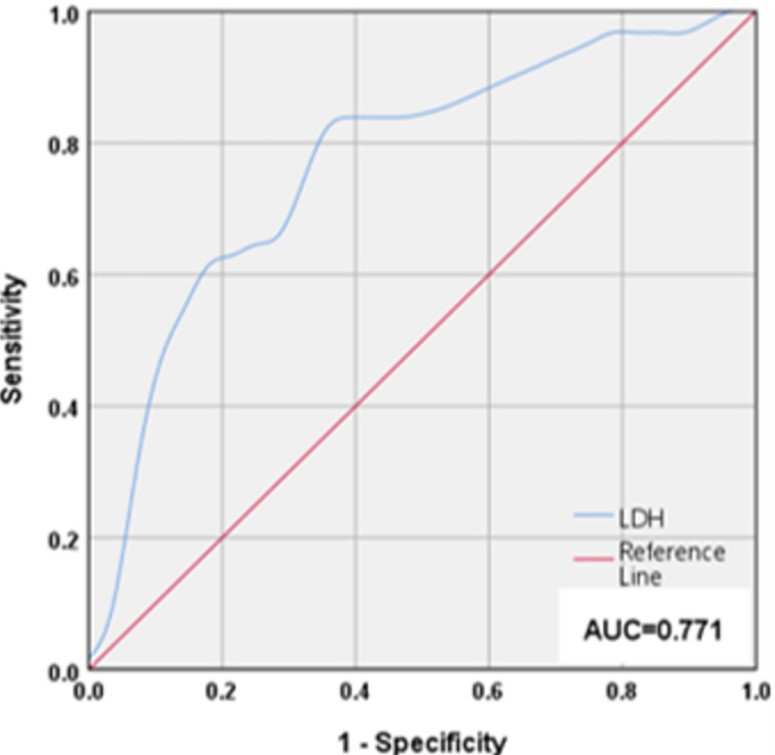

**Fig 2. Receiver operating characteristic curves for LDH for screening patients with severe polytrauma for the presence of chest injury.** This is an ROC curve figure with LDH as a variable using sever polytrauma patients with concomitant chest injuries as a positive result and polytrauma patients without concomitant chest injuries as a negative result. AUC (= 0.771, p<0.001) provides a measure of the overall diagnostic accuracy of LDH in identifying sever polytrauma patients with chest injurie. LDH: Lactate dehydrogenase. AUC: Area Under Curve.

elevated LDH levels were more frequently observed in patients with fractures of the chest wall (6/27 vs. 24/37, p = 0.001), pulmonary contusion (0/27 vs. 18/37, p<0.001), and cardiovascular diseases (0/27 vs. 6/37, p = 0.035) (Table 5).

## Discussion

This study aimed to find clinical indicators that could quickly and accurately screen patients with PT for the presence of chest injury using multifactorial analysis modeling. The thoracic

**Table 4. Sensitivity, specificity, PPV and NPV of LDH for screening patients with severe polytrauma for the presence of chest injury.**

| Variable | LDH>365U/l |
|---|---|
| Sensitivity | 80.65% |
| Specificity | 67.86% |
| PPV | 73.53% |
| NPV | 76.00% |

This table shows the sensitivity, specificity, positive predictive value, and negative predictive value for predicting the presence or absence of comorbid chest injuries in patients with severe polytrauma using LDH = 365 U/L as the threshold value.

LDH: Lactate dehydrogenase. PPV: positive predictive value; NPV: negative predictive value.

**Table 5. Comparison of diagnostic results of chest injuries in patients with different LDH.**

| Diagnosis of chest injuries | LDH<365 U/L n = 27 | LDH≥365 U/L n = 37 | P value |
|---|---|---|---|
| Fractures of the chest wall: ribs, sternum, thoracic vertebrae | 6 | 24 | 0.001** |
| Extra-thoracic wall fractures: scapula, clavicle | 0 | 3 | 0.257 |
| Pulmonary contusion | 0 | 18 | <0.001** |
| Pneumothorax, haemothorax, pleural effusion | 4 | 13 | 0.089 |
| Pneumonia | 1 | 7 | 0.124 |
| Cardiovascular diseases: aortic dissection, myocardial contusion, pericardial effusion, mediastinal haematoma | 0 | 6 | 0.035* |
| Blunt aortic injury | 0 | 2 | 0.504 |
| Myocardial contusion | 0 | 2 | 0.504 |
| Pericardial effusion, mediastinal haematoma | 0 | 3 | 0.257 |

This table shows the distribution of different chest injuries in different LDH patients.

Results are shown in the number of patients.

*: P-value<0.05;

**: P-value<0.001.

cavity contains life-sustaining organs, such as the heart and lungs, and chest injuries are of particular concern in the assessment and treatment of patients with PT [13].

LDH is an enzyme found in many body tissues, including the heart, lungs, liver, kidneys, skeletal muscle, and blood cells. When cells are damaged by injury, LDH is released into the bloodstream. Therefore, systemically elevated levels of LDH might be seen as an indicator of tissue damage [14]. The physiological role of LDH is to catalyze the reversible reaction of pyruvate with lactate [15] and previous studies have already shown that lactate, the substrate of LDH, can be an excellent prognostic indicator for trauma patients [1, 16, 17]. Against this background, Régnier, M.A., et al. [18] showed that elevated levels of lactate were strongly correlated with trauma severity. Furthermore, Park, H.O., et al. [1] and Parsikia, A., et al [16]. found that elevated lactate concentrations were associated with post-traumatic respiratory failure, and death. Data from the present study revealed that lactate, however, does not identify site-specific injuries, specifically chest injuries, in patients with PT. This may be because lactate, as a biomarker of inadequate tissue perfusion, is highly correlated with the overall burden of trauma in PT patients. In addition, the included patients presented with a very high ISS compared to other studies that investigated the role of lactate in PT [1, 7, 16, 19]. However, the present study demonstrated that serum LDH levels exhibited a correlation with the occurrence of concomitant chest injuries among patients afflicted with PT. LDH has been used as an aid in the diagnosis of many diseases, such as cancer, thyroid disease, and tuberculosis [8], and its diagnostic value for lung injury caused by 2019-nCoV has received widespread attention [9, 20]. LDH levels may also rise due to injuries to organs, such as the liver and kidneys [21, 22]. However, elevations in LDH levels have been observed in a broad spectrum of chest injuries, including those affecting the lungs and myocardium [14, 23, 24]. Thus, LDH could serve as a viable biomarker for the identification of individuals suffering from PT with concomitant chest injuries. As the concentration of LDH in tissues is approximately 500-fold greater than that in serum [25], LDH from the affected tissues is liberated into the bloodstream, culminating in an increase in serum LDH levels. This mechanism might even be enhanced by the fact that thoracic organs are very susceptible to hypoperfusion and hypoxemia, thereby precipitating more widespread tissue damage and an augmented serum LDH level. Although, the ubiquitous elevation in serum LDH levels is frequently regarded as lacking tissue specificity, none

of the other organ specific laboratory results within this study cohort presented with significant changes [26]. Thus, this study demonstrated that elevated LDH levels at the time of admission in PT may serve as a reliable indicator of chest injury, a finding which has not been previously reported in the literature. The implications of these findings are significant for enhancing clinical diagnosis and treatment strategies. As a consequence of the presented findings, additional investigation of the thoracic organs in PT with increased LDH could be promptly initiated [13, 27]. The finding that LDH directly correlates with chest injury patterns opens new possibilities for small hospitals that may not have direct access to high-resolution CT scanners in their shock rooms. This could also be an on-scene option using a "lab on a chip" [28], so that blood samples could be evaluated before arriving at a hospital or in situations where the patient cannot be transported (e.g., natural disasters or battlefields). In addition, it might reduce the overall costs associated with this diagnosis.

Additionally, through an analysis of the diagnoses of patients with varying LDH levels, we identified that fractures of the chest wall, pulmonary contusions, and cardiovascular diseases may contribute to elevated LDH. This finding suggests a potential direction for subsequent diagnostic tests in polytrauma patients with concomitant chest injuries as identified by elevated LDH levels. It is important to note that due to the small sample size in this study and the low incidence of certain injuries (e.g., blunt aortic injury, myocardial contusion), it is challenging to conduct specific analyses for each individual condition. Therefore, we do not advocate for LDH to completely replace other investigations such as ultrasound, CT, etc., as they are necessary to confirm the diagnosis and clarify the progression of the disease. Nonetheless, our current results provide valuable insights into the direction of further examinations, which can aid in optimizing the diagnostic process and allocating medical resources more effectively.

The fact that a wide variety of easily available and inexpensive methods exist for the detection of LDH (e.g., colorimetric, spectrophotometric, and fluorescence methods) confers further attractiveness on LDH as a marker to use in the fast diagnosis of chest injury in patients with polytrauma [29]. Combined with the predictive model of this study, it provides a convenient and inexpensive new method for the diagnosis of patients with multiple injuries accompanied by chest damage.

## Conclusion

LDH may be a promising indicator for screening for the presence of chest injury in patients with polytrauma. Although the presented findings delineate an association between LDH levels and PT accompanied by chest injuries, the precise underlying mechanism warrants additional exploration.

## Strengths & limitations

Our study is characterized by several notable strengths: the trauma severity of the subjects investigated was comparably high and the comprehensiveness of our patient data is significantly enhanced by the advanced degree of digital integration within our clinical operations, thereby enabling more exhaustive analyses. Yet, this is a single-center, small sample study with a retrospective design, which may introduce selection bias. Due to the limited sample size, subgroup analyses for specific chest injuries like lung contusions and rib fractures were not feasible. Consequently, while our study's capacity to advance treatment modalities is restricted, it proves invaluable for early diagnosis and triage of patients.

Considering the variety of comorbidities that could influence the levels of LDH in patients, further research is imperative. The objective of such studies would be to discern additional comorbidities impacting LDH levels, thereby facilitating the classification of patients with

severe polytrauma into subgroups based on their specific comorbid conditions. This stratification would enable more precise evaluation of the efficacy of LDH in detecting the occurrence of chest injuries within this patient cohort.

## Supporting information

**S1 Table. All results of comparison of biochemical markers between patients with severe polytrauma without chest injury (Group A) and with chest injury (Group B).** This table shows the results of the comparison between Group A and Group B for all biomarkers. (DOCX)

**S2 Table. All results of the correlation analysis of various biochemical parameters with chest injury in polytrauma patients.** This table delineates the findings from an analysis examining the correlation of all variables with the presence or absence of concurrent chest injuries in individuals suffering from severe polytrauma, wherein the presence of such injuries is denoted as a positive outcome. (DOCX)

## Acknowledgments

The authors would like to express their appreciation to the medical and nursing teams of Department of Orthopaedics, Trauma and Reconstructive Surgery, University Hospital RWTH Aachen, who helped with this study. We would like to mention the convenient service of the proof-reading company Scribendi.

## Author Contributions

**Conceptualization:** Weining Yan, Felix Bläsius, Johannes Greven, Klemens Horst.

**Data curation:** Weining Yan.

**Formal analysis:** Weining Yan.

**Investigation:** Tabea Wahl.

**Methodology:** Weining Yan, Felix Bläsius, Johannes Greven, Klemens Horst.

**Software:** Weining Yan, Tabea Wahl.

**Supervision:** Frank Hildebrand, Elizabeth Rosado Balmayor, Johannes Greven.

**Validation:** Weining Yan.

**Visualization:** Weining Yan.

**Writing – original draft:** Weining Yan.

**Writing – review & editing:** Weining Yan, Elizabeth Rosado Balmayor, Johannes Greven, Klemens Horst.

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
