## [Decision Letter · Decision Letter 0]

13 May 2024

PONE-D-24-08209Lactate dehydrogenase can be used for differential diagnosis to identify patients with severe polytrauma with or without chest injury—a retrospective studyPLOS ONE

Dear Dr. Yan,

Thank you for submitting your manuscript to PLOS ONE. After careful consideration, we feel that it has merit but does not fully meet PLOS ONE’s publication criteria as it currently stands. Therefore, we invite you to submit a revised version of the manuscript that addresses the points raised during the review process.

We look forward to receiving your revised manuscript.

Kind regards,

Zsolt J. Balogh, MD, PhD, FRACS

Academic Editor

PLOS ONE

Journal Requirements:

**Additional Editor Comments:**

Dear Authors,

Our reviewers identified some major concerns related to your submission.

I would like to provide the opportunity to address these and have a chance for them to reconsider their position.

Reviewers' comments:

Reviewer's Responses to Questions

**Comments to the Author**

1. Is the manuscript technically sound, and do the data support the conclusions?

Reviewer #1: No

Reviewer #2: No

2. Has the statistical analysis been performed appropriately and rigorously? 

Reviewer #1: I Don't Know

Reviewer #2: I Don't Know

3. Have the authors made all data underlying the findings in their manuscript fully available?

Reviewer #1: Yes

Reviewer #2: Yes

4. Is the manuscript presented in an intelligible fashion and written in standard English?

Reviewer #1: Yes

Reviewer #2: Yes

5. Review Comments to the Author

Reviewer #1: The authors have presented a well written and reasoned article outlining the clinical relevance and

implication of their hypothesis and results. There are some issues that I think would benefit from

addressing to clarify the results and utilise current validated definitions.

Issues

1. The authors use ISS>16 as a definition for polytrauma, without justification. This justification

is required as it is not consistent with the current validated “Newcastle definition”. Two

options are suggested to address this:

a. Change the inclusion criteria to match the validated “Newcastle definition” (AIS > 2

in 2 body regions)

or

b. Change the nomenclature to “multiple trauma” or “major trauma” depending on the

number of body regions involved

2. The authors use the term “severe polytrauma” when describing patients with chest injury,

but do not clarify what constitutes “severe polytrauma”.

3. The threshold for patients having a chest injury is not articulated – was it AIS ≥1, another

cutoff, or based on clinical/radiological findings.

4. Whether the patient groups are matched in other variables, injuries and demographics was

not outlined. With a small sample size there are many possible confounders. For example the

ISS or presence of solid organ injury were not included. Such confounding factors need to be

addressed.

Reviewer #2: The authors present an interesting set of data, albeit small.

The structures actually injured in the chest are not described.

they do not describe how to use these data to change treatment.

i would add CXR to the modalities available to quickly determine cheat injury.

6. PLOS authors have the option to publish the peer review history of their article (what does this mean?). If published, this will include your full peer review and any attached files.

Reviewer #1: No

Reviewer #2: No

---

## [Author Response · Author response to Decision Letter 0]

21 May 2024

Dear Editor and Reviewers,

Thank you very much for your time and effort in reviewing our manuscript titled " Lactate dehydrogenase can be used for differential diagnosis to identify patients with severe polytrauma with or without chest injury—a retrospective study" We greatly appreciate your valuable comments and suggestions. The feedback provided has been instrumental in enhancing the quality and clarity of our manuscript. We have carefully considered each point raised and have made corresponding revisions to address these concerns.

We are resubmitting our manuscript after incorporating the changes suggested by you and hope that the revised manuscript meets the esteemed standards of PlosOne. Below, we provide detailed responses to each of the comments and outline the modifications we have made to the manuscript. Thank you once again for the opportunity to improve our work.

Response to Reviewer #1

1. In response to the first and second comments. Thank you for your suggestion, as you said, there is a descriptive problem with the definitions here, now we have defined polytrauma, severe trauma and severe polytrauma more accurately based on your suggestions and have added references to support the basis of our definitions. Changes are made on lines 79 to 83 of the manuscript.

2. In response to the third comment. Yes, the threshold for patients having a chest injury is chest AIS ≥1. Thank you very much for the reminder, we have added a description on line 83 of the manuscript.

3. In response to the fourth comment. 

In this study, we analyzed patients by indicators such as age, gender, and Injury Severity Score (ISS), observing no significant differences in outcomes among the groups; detailed results are provided in the Supporting Materials. Our patient cohort consisted of individuals suffering from severe polytrauma. We categorized these patients based on the presence or absence of chest injuries. It is important to note that while other parenchymal organ injuries such as liver, kidney, and spleen were identified, their incidence did not differ between our two groups of patients (6/29 vs. 8/35, P=0.835). Our findings revealed no differences in ISS scores between groups, indicating a uniform severity of trauma across the cohort. This suggests that the presence of chest injuries does not correlate with a higher overall injury severity under the conditions of this study.

Moreover, we addressed potential confounding factors using a multifactorial logistic regression model. This analysis helped to clarify the impact of variables that appeared significant in univariate analyses, such as glutamic-pyruvic transaminase (GPT) and lactate levels, which ultimately did not affect the outcomes related to the diagnosis of chest injuries. Our results highlight that lactate dehydrogenase (LDH) levels at admission are significantly associated with the differential diagnosis of chest injuries in cases of severe polytrauma.

Given the limited sample size of our study, it was not feasible to perform subgroup analyses for multiple organ injuries comprehensively. Therefore, our laboratory is committed to expanding the sample size in future studies. This expansion will allow us to explore additional biomarkers and potentially introduce new screening indicators for other organ injuries. We trust that our rigorous approach to eliminating confounding factors meets your standards and contributes to the robustness of our findings.

Response to Reviewer #2

1. Thank you very much for your insightful comments and for highlighting the limitations of our study. In our analysis, the categorization of chest injuries included various conditions such as lung contusion, rib fractures, sternal fractures, pneumothorax, among others. Due to the limited sample size, detailed subgroup analyses within these categories were not feasible. Consequently, we grouped these conditions under a general category of "chest injuries," defined as chest AIS (Abbreviated Injury Scale) ≥ 1.

As a result, our study primarily demonstrates the utility of lactate dehydrogenase (LDH) as a diagnostic biomarker for detecting chest injuries in patients with severe polytrauma. We acknowledge, as you pointed out, that this classification approach limits our study’s implications for enhancing specific treatment strategies for distinct injuries such as rib fractures or lung contusions. However, the accessibility and rapidity of LDH testing can significantly aid prehospital diagnosis. This is particularly beneficial in scenarios where immediate access to well-equipped medical facilities is not possible or during large public events that necessitate quick triage of patients.

We have also taken your advice into consideration and have included a detailed description of these limitations in the Limitations section of our manuscript. We appreciate your rigorous review and valuable suggestions, which have undoubtedly improved the manuscript.

2. Thank you very much for your suggestion regarding the inclusion of chest X-ray (CXR) as part of the screening criteria. Indeed, in our study, CXR was utilized as one of the foundational diagnostic tools for patients with chest injuries. We acknowledge that incorporating CXR into the screening protocol can enhance the accuracy of injury detection.

However, the primary focus of our study was to evaluate the effectiveness of a rapid hematological screening method as a standalone diagnostic tool, particularly in scenarios where radiological facilities, such as CXR, are not readily accessible—such as at the scene of a natural disaster. The intent was to provide a feasible alternative for initial assessment in severe polytrauma patients, both with and without apparent chest injuries, when traditional radiological diagnostics are unavailable.

Therefore, while CXR results were considered in the patient assessment, our manuscript emphasizes the utility and applicability of hematological screening in resource-limited settings. We believe this focus is crucial for advancing pre-hospital care in less equipped environments, and it complements existing diagnostic protocols that rely on imaging technologies.

We are deeply grateful for the detailed and constructive feedback provided by the reviewers and the editor. Your insights have significantly contributed to refining our manuscript and ensuring that it meets the rigorous standards of PlosOne. We have endeavored to address all the points raised in a comprehensive manner and have made revisions that we believe enhance the value and clarity of our work.

We hope that the changes we have implemented are satisfactory and meet the expectations of the review committee. We are committed to making any further modifications if needed and are looking forward to your suggestions and final decision.

Thank you once again for your guidance and support throughout the review process. We appreciate the opportunity to contribute to PlosOne and eagerly anticipate the possibility of our work being published.

Sincerely yours,

Weining Yan

---

## [Decision Letter · Decision Letter 1]

14 Jun 2024

PONE-D-24-08209R1Lactate dehydrogenase can be used for differential diagnosis to identify patients with severe polytrauma with or without chest injury—a retrospective studyPLOS ONE

Dear Dr. Yan,

Thank you for submitting your manuscript to PLOS ONE. After careful consideration, we feel that it has merit but does not fully meet PLOS ONE’s publication criteria as it currently stands. Therefore, we invite you to submit a revised version of the manuscript that addresses the points raised during the review process.

We look forward to receiving your revised manuscript.

Kind regards,

Zsolt J. Balogh, MD, PhD, FRACS

Academic Editor

PLOS ONE

Reviewers' comments:

Reviewer's Responses to Questions

**Comments to the Author**

1. If the authors have adequately addressed your comments raised in a previous round of review and you feel that this manuscript is now acceptable for publication, you may indicate that here to bypass the “Comments to the Author” section, enter your conflict of interest statement in the “Confidential to Editor” section, and submit your "Accept" recommendation.

Reviewer #1: (No Response)

Reviewer #2: All comments have been addressed

2. Is the manuscript technically sound, and do the data support the conclusions?

Reviewer #1: Partly

Reviewer #2: No

3. Has the statistical analysis been performed appropriately and rigorously? 

Reviewer #1: Yes

Reviewer #2: Yes

4. Have the authors made all data underlying the findings in their manuscript fully available?

Reviewer #1: Yes

Reviewer #2: Yes

5. Is the manuscript presented in an intelligible fashion and written in standard English?

Reviewer #1: (No Response)

Reviewer #2: Yes

6. Review Comments to the Author

Reviewer #1: Thanks for the responses and clarifications from the authors.

I have one further suggestion. I do not understand the distinction between "severe polytrauma" and "polytrauma" and suggest removing it from the article. By the 2xAIS>2 definition now used by the authors, the minimum ISS is 18. Therefore the distinction of severe polytrauma as 2xAIS>2 + "severe trauma" as ISS of ≥16 shouldn't be required.

Reviewer #2: the authors have addressed the reviewers comments within the limitations of their very small dataset. I remain unconvinced that LDH is a useful tool in this situation. the authors need to describe what chest injuries injuries cause this elevation. Rib fractures do not need immediate treatment, neither does a pulmonary contusion. what about pericardial tamponade? myocardial contusion? pneumothorax?

this is a a very preliminary finding that should be explored to see what injury its associated with.

why wouldn't a pulse oximeter and US just be as useful in a mass casualty situation?

7. PLOS authors have the option to publish the peer review history of their article (what does this mean?). If published, this will include your full peer review and any attached files.

Reviewer #1: No

Reviewer #2: No

---

## [Author Response · Author response to Decision Letter 1]

20 Jun 2024

Dear Editor and Reviewers,

We sincerely appreciate the time and effort you have invested in reviewing our manuscript titled "Lactate dehydrogenase can be used for differential diagnosis to identify patients with severe polytrauma with or without chest injury—a retrospective study." Your insightful comments and constructive feedback have been invaluable in refining and improving the quality of our work.

Following your recommendations from the initial review, we have undertaken a thorough revision of our manuscript. Your detailed and thoughtful suggestions have significantly contributed to the enhancement of the clarity, robustness, and overall scientific merit of our study. We are deeply grateful for your guidance.

In this second round of revisions, we have meticulously addressed each of the points you raised. We believe the changes we have implemented strengthen the manuscript and align it more closely with the high standards of PlosOne. We have summarized our responses to each comment below, detailing the modifications made to the manuscript to address your concerns.

Thank you once again for providing us with this invaluable opportunity to further improve our work. We hope that the revised manuscript meets your expectations and the esteemed standards of PlosOne.

Response to Reviewer #1

Thank you very much for your insightful comments. As you rightly pointed out, the original description was overly cumbersome. In accordance with your suggestion, I have retained only the definition of 'Severe polytrauma' and have revised the description of the inclusion criteria (lines 66, 79-81). I trust these modifications address your concerns and enhance the clarity and precision of the manuscript.

Response to Reviewer #2

1. Thank you for your constructive feedback regarding the need for a deeper analysis of the sources of elevated LDH levels, which could indeed enhance the credibility and practical value of our study. In response to your suggestion, we conducted a detailed examination of the specific injuries contributing to elevated LDH levels. Given the complexity of these diagnoses, we categorized the injuries as follows: fractures of the chest wall (including ribs, sternum, and thoracic vertebrae), extra-thoracic wall fractures (scapula and clavicle), pulmonary contusion, pleural diseases (pneumothorax, haemothorax, pleural effusion), pneumonia, and cardiovascular diseases (aortic dissection, myocardial contusion, pericardial effusion, mediastinal haematoma).

Our results indicate significant associations between elevated LDH levels and the following conditions: fractures of the chest wall (p=0.001), pulmonary contusion (p<0.001), and cardiovascular diseases (p=0.035). These findings provide valuable guidelines for directing further diagnostic examinations in patients.

In our manuscript, we have elaborated on the significance of these findings and discussed the limitations arising from the low incidence of certain conditions, such as myocardial contusion and aortic coarctation, which prevented a more detailed disease-specific analysis. These changes have been incorporated into lines 198-211 and 260-269 of the revised manuscript. I trust these amendments address your concerns and contribute positively to the manuscript.

2. Thank you for your suggestion regarding the use of pulse oximetry and ultrasound in our study. Our investigation did include an analysis of the partial pressure of blood oxygen; however, we found that it does not reliably indicate chest injuries. This may be due to its insensitivity to injuries that do not directly impact ventilation and gas exchange, noting that chest injuries can extend beyond the lungs and airways.

While we agree that comprehensive diagnostic tools like ultrasound or X-ray can enhance diagnostic accuracy, the primary goal of our study was to develop a rapid screening method for chest injuries in polytrauma patients. This approach is intended to streamline patient triage and optimize the allocation of healthcare resources. LDH testing, which can be performed concurrently with routine blood gas analyses, offers a quick and readily available diagnostic measure without the need for additional equipment or specialized personnel such as ultrasonographers.

We have detailed the benefits of using LDH as a diagnostic tool in the sections on lines 48-53, 252-258, and 266-275 of our manuscript. These sections explain how LDH integrates into clinical protocols to provide effective, efficient care in emergency settings. I appreciate your comments and hope that the revisions and clarifications meet the article's aims and your expectations.

3. Thank you for your insightful comments. As you noted, our study is a preliminary investigation with a limited sample size. Despite these constraints, our methodology—which includes controlling for confounders—has successfully demonstrated a rapid screening technique for chest injuries. This method also aims to guide clinical examinations and optimize the allocation of healthcare resources effectively.

However, we acknowledge several limitations, particularly concerning the specificity of our screening for individual organs and distinct pathologies. In response to these limitations and guided by your valuable suggestions, our laboratory is committed to expanding the sample size in future studies. This expansion will enable us to explore additional biomarkers and potentially develop new screening metrics for other organ injuries.

We appreciate your guidance, which has been instrumental in refining our research approach. We trust that the revisions and future study directions outlined here meet your expectations and contribute meaningfully to the field.

We extend our deepest gratitude to the reviewers and the editor for their detailed and constructive feedback. Your insights have been instrumental in refining our manuscript and ensuring it adheres to the rigorous standards of PlosOne. We have diligently addressed all the points raised, implementing revisions that we believe significantly enhance the clarity, robustness, and overall value of our work.

We sincerely hope that the changes we have made meet your expectations and are satisfactory to the review committee. We remain fully committed to making any further modifications if needed and greatly value your continued guidance and support.

Thank you once again for this opportunity to improve our manuscript. We appreciate the chance to contribute to PlosOne and eagerly anticipate the possibility of our work being published.

Sincerely yours,

Weining Yan

---

## [Editor Report · Decision Letter 2]

24 Jun 2024

PONE-D-24-08209R2Lactate dehydrogenase can be used for differential diagnosis to identify patients with severe polytrauma with or without chest injury—a retrospective studyPLOS ONE

Dear Dr. Yan,

Thank you for submitting your manuscript to PLOS ONE. After careful consideration, we feel that it has merit but does not fully meet PLOS ONE’s publication criteria as it currently stands. Therefore, we invite you to submit a revised version of the manuscript that addresses the points raised during the review process.

We look forward to receiving your revised manuscript.

Kind regards,

Zsolt J. Balogh, MD, PhD, FRACS

Academic Editor

PLOS ONE

Additional Editor Comments:

Dear Dr. Yan,

I appreciate your revision efforts and providing more information.

One of our reviewers still has major concerns and I have some questions to clarify:

- You need to rationalise pragmatism for using LDH as a diagnostic tool in acute trauma. What are you diagnosing with it. The clinical examination within minutes identifies if we need to decompress the chest or we need to do ED thoracotomy or to start haemostatic resuscitation with coagulation factors and packed red blood cells. in 3 minutes we have a chest x-ray, which excludes almost all immediately life-threatening conditions in the chest and raises suspicion for others. We have CT scan with contrast within 15 minutes in any patients who had an injury mechanism, which can cause major chest injury and had polytrauma. What additional benefit LDH add during this early phase of rapid work-up? I doubt that it would prevent some imaging and I would doubt that it would indicate earlier intervention than based on clinical exam, chest x-ray and panscan.

- Aortic dissection is a medical condition not traumatic, I assume you are referring to torn thoracic aorta or in other words blunt aortic injury.

- LDH is a cell necrosis marker (that's it), to make your conclusions you need to convince that it is a better cell necrosis marker than others and also specific to chest injuries

- Your study does not include an essential chest (cardiac) injury marker, which is routinely used in most trauma centres: Troponin. Is LDH better than troponin? is it more specific? I doubt.

- the ~75% NP and PPV is not stellar, I am sure clinical exam/chest x-ray/CT and overall pattern recognition by clinicians will beat it.

- Please address the potential of Type2 error in your manuscript from the fact that you evaluate a bunch of variables and by chance one popping up just significant.

- Please provide convincing information that LDH is specific to chest injury and not just a marker of overall severe tissue disruption.

- your analysis does not address the treatment factors, which can have an effect on LDH (resuscitation blood products and surgical interventions).

- I am sure you agree that LDH is unlikely to be a single marker for diagnosing chest injury, could you please elaborate how much better is it with addition to our existing ones? You do not have ROCs for clinical exam, chest x-ray and CT scan or echocardiography, ECG etc.

I appreciate your work on this manuscript and its revisions. My questions are merely pragmatic clarifications as we have not find yet a single laboratory test in trauma care identifying region specific injuries and the current data available about LDH is not convincing to me based on 64 patients.

---

## [Author Response · Author response to Decision Letter 2]

27 Jun 2024

Dear Reviewers and Editor,

We would like to express our sincere gratitude for the continued time and effort you have dedicated to reviewing our manuscript titled "Lactate dehydrogenase can be used for differential diagnosis to identify patients with severe polytrauma with or without chest injury—a retrospective study." Your ongoing insightful comments and valuable suggestions have been crucial in further refining and enhancing the quality of our work.

Following your detailed feedback from the second review, we have once again undertaken a revision of our manuscript. We greatly appreciate the constructive guidance provided, which has significantly contributed to the robustness and clarity of our study. Each point raised has been carefully considered, and corresponding revisions have been made to address these concerns thoroughly.

We are resubmitting our manuscript after incorporating these additional changes and hope that the revised version meets the high standards of PlosOne. If you believe that incorporating our detailed responses directly into the body of the manuscript would be beneficial, we would be more than happy to do so. Below, we provide detailed responses to each of the comments and outline the modifications made to the manuscript. Thank you once again for this invaluable opportunity to improve our work.

Responses to Reviewer Comments

1. Rationalization of LDH as a Diagnostic Tool in Acute Trauma

Thank you for your detailed and insightful comments regarding the limitations of LDH as a diagnostic reference marker in well-resourced medical environments. We fully acknowledge that our study has limited utility in guiding the diagnosis and treatment of specific diseases under such conditions. However, the primary aim of our research was to facilitate the rapid screening of polytrauma patients for chest injuries, thereby providing a basis for the necessity of further chest examinations.

In scenarios where medical resources are severely constrained—such as during natural disasters (earthquakes, tsunamis), conflicts, or widespread epidemics—conducting comprehensive examinations on all potentially injured patients within a short timeframe is unfeasible. In these situations, any tool that aids in patient triage can significantly improve the allocation of medical resources and potentially save lives. Thus, our study highlights the utility of LDH in identifying patients with potential chest injuries, allowing healthcare providers to prioritize their evaluations more effectively during periods of resource scarcity, as detailed in lines 252-258 of the manuscript.

Regarding your concern that the use of LDH may deter some imaging tests, we believe our findings will instead assist physicians in determining when imaging is warranted. For instance, a patient with an LDH level exceeding 365 U/L would prompt increased vigilance for chest injuries, thereby reducing the risk of missed diagnoses, even if the primary injury is elsewhere, such as an open fracture of the lower extremity with hemorrhagic shock.

Moreover, our study aimed to identify patients who are "more likely" to have chest injuries, allowing for prioritized treatment and avoiding unnecessary delays for those less likely to have such injuries. This stratification is crucial for the rational allocation of limited healthcare resources, enabling more efficient and targeted patient care.

We hope this clarifies the scope and significance of our study in contexts with limited medical resources.

2. Clarification on Aortic Dissection

Thank you for your correction. We indeed referred to aortic tears resulting from trauma. We have updated the description in the manuscript to "blunt aortic injury."

3. LDH as a Cell Necrosis Marker

Thank you for your insightful comments. Lactate dehydrogenase (LDH) is recognized as a marker of cellular necrosis, and our findings suggest that it can be effectively used for the differential diagnosis of chest injuries in polytrauma patients. However, we did not evaluate LDH in comparison to other markers of cellular necrosis, such as TNFα, because these are not commonly assessed in routine clinical practice. This approach allows our results to be more directly applicable to everyday clinical settings.

While it is acknowledged that LDH lacks tissue specificity, displaying only a 1.5-fold difference in activity between tissues with the highest (e.g., liver) and lowest (e.g., kidney) LDH levels, our study still observed elevated LDH levels in patients with chest injuries, despite having similar overall trauma severity scores (ISS=30.43 ± 9.28 vs. 30.88 ± 10.24, P=0.894). This elevation likely reflects the involvement of organs within the chest that are critical to respiratory and circulatory functions, resulting in greater cellular necrosis and subsequent release of LDH into the bloodstream. 

We have detailed this rationale in lines 237-252 of the manuscript. If LDH can demonstrate significant differences between groups, the utility of other markers, even if potentially more specific, becomes less critical in the context of routine clinical assessments. We hope this clarification meets your requirements and enhances the manuscript.

4. Comparison with Troponin

Thank you for your insights regarding the use of cardiac troponins (cTnI and cTnT) as specific markers of myocardial injury. We recognize that troponins are superior to lactate dehydrogenase (LDH) in identifying cardiac damage. However, the primary diagnostic aim of our study was not solely to detect cardiac injury but to ascertain the presence of any chest injury in polytrauma patients. Since the chest includes a variety of tissues, such as the lungs and chest wall, normal troponin levels do not necessarily exclude the presence of non-cardiac chest injuries. Our results support the utility of LDH as a broader marker in this context, providing a simple and routinely available diagnostic option that can indicate a range of chest injuries during initial patient evaluations.

Additionally, we also analyzed creatine kinase (CK) and CK-MB in our study. Despite the high cardiac specificity of CK-MB, it did not show potential in identifying general chest injuries among polytrauma patients, further highlighting LDH’s broader applicability for this purpose. These findings reinforce the importance of LDH in the initial assessment of polytrauma patients, where a comprehensive evaluation of all possible chest injuries is critical.

We trust this explanation clarifies the rationale behind our focus on LDH and its role in our study. We appreciate your valuable feedback and hope that our findings contribute meaningfully to the field.

5. Positive and Negative Predictive Values

Thank you for your pertinent and insightful comments. It is generally accepted that NPV and PPV values above 80% are ideal for clinical diagnosis. While our results are very close to this threshold, we acknowledge that the diagnosis of specific chest injuries ultimately relies on targeted tests such as ultrasound, CT, and other imaging modalities.

The significance of our study lies in the potential of LDH to aid physicians in pre-screening patients who are likely to have chest injuries. This preliminary screening can guide clinicians to perform more targeted and efficient examinations, optimizing the use of diagnostic resources and potentially improving patient outcomes.

6. Addressing Type 2 Error

Thank you for your attention to the details of our study. We validated our data through efficacy analysis. In our binary logistic regression analysis, we achieved an acceptable result with N=64, α=0.05, and a power of 0.83 (a power of 0.8 is usually acceptable in scientific research). We have prepared the test report in PDF format and are ready to provide it for your review. Please advise on the preferred method for submission if submission is required, as the manuscript submission guidelines do not specify where such documents should be uploaded or sent.

7. Specificity to Chest Injury

As discussed in response to question No. 3, our statistical analysis clearly demonstrates that elevated LDH levels correlate significantly with additional chest trauma in severely injured patients. This correlation is mathematically significant, reinforcing LDH's utility as a biomarker in this context.

Although LDH is not tissue-specific—studies indicate only about a 1.5-fold difference between the tissues with the highest LDH values (e.g., liver) and those with the lowest (e.g., kidney)—we observed a notable increase in LDH levels in patients with chest injuries, despite comparable overall trauma levels (ISS=30.43 ± 9.28 vs. 30.88 ± 10.24, P=0.894). This observation may be attributed to the involvement of organs within the chest that are essential to the respiratory and circulatory systems, which could lead to increased cellular necrosis and subsequent LDH release into the bloodstream, as detailed in lines 237-252 of the manuscript.

It is important to recognize that elevated LDH levels in patients with chest injuries may not solely arise from the factors described. The underlying mechanisms remain a subject for further investigation, a direction we have committed to pursuing in future research, as outlined in our article.

8. Influence of Treatment Factors on LDH

Thank you for your suggestion. In our analysis, we considered the effect of blood transfusion by examining the transfusion of red cells, plasma, and platelets in patients with and without chest injuries. Our results, presented in S1 Table, indicate no significant difference between the groups.

Surgery may indeed affect LDH levels, which could explain why the between-group differences in LDH became non-significant at subsequent observation time points. However, our findings demonstrate that LDH levels at admission can be used to identify the presence of chest injury in polytrauma patients. At the time of admission, none of the patients had undergone surgery, so the LDH levels were not influenced by surgical intervention.

9. Addition to Existing Diagnostic Methods

Thank you for your insights regarding the reliability of using elevated LDH levels as a sole diagnostic marker. I concur that a definitive diagnosis of specific diseases based solely on LDH levels would be unreliable. However, our findings suggest that LDH can serve effectively as an indicative marker for potential concomitant chest injuries in polytrauma patients. It is important to clarify that our study does not advocate for the abandonment of established diagnostic tests such as X-rays, CT scans, and ultrasounds; rather, we propose that LDH can be used to guide more targeted diagnostic efforts.

To prevent any potential misunderstandings, we have included a clarifying note in lines 266-268 of the manuscript, emphasizing that LDH should not replace more comprehensive diagnostic methods like ultrasound or CT scans. These tools are indispensable for confirming disease diagnoses and monitoring disease progression.

In resource-constrained settings, such as during natural disasters or conflicts, the ability to quickly classify patients using basic laboratory tests like LDH can be invaluable. This method enables a more efficient allocation of limited medical resources by prioritizing the examination of patients most likely to have significant injuries. Additionally, detecting elevated LDH levels can raise awareness among physicians of possible underlying injuries that may not be immediately apparent from the patient's primary complaints.

We hope this explanation addresses your concerns and clarifies the intent and scope of our research.

We are deeply grateful for the detailed and constructive feedback provided by the reviewers and the editor throughout this review process. Your insights have been instrumental in refining our manuscript and ensuring it meets the rigorous standards of PlosOne. We have endeavored to address all the points raised in a comprehensive manner, and we believe the revisions we have implemented significantly enhance the value and clarity of our work.

We sincerely hope that the changes made are satisfactory and meet the expectations of the review committee. Should you find it advantageous for our responses to be integrated into the manuscript itself, we are more than willing to make such modifications. We remain committed to making any further adjustments if needed and greatly value your continued guidance and support.

Thank you once again for your assistance and for providing us with this opportunity to improve our manuscript. We appreciate the chance to contribute to PlosOne and eagerly anticipate the possibility of our work being published.

Sincerely yours,

Weining Yan

---

## [Editor Report · Decision Letter 3]

19 Jul 2024

Lactate dehydrogenase can be used for differential diagnosis to identify patients with severe polytrauma with or without chest injury—a retrospective study

PONE-D-24-08209R3

Dear Dr. Yan,

We’re pleased to inform you that your manuscript has been judged scientifically suitable for publication and will be formally accepted for publication once it meets all outstanding technical requirements.

Kind regards,

Zsolt J. Balogh, MD, PhD, FRACS

Academic Editor

PLOS ONE

Additional Editor Comments (optional):

Thank you.
---

## [Editor Report · Acceptance letter]

23 Jul 2024

PONE-D-24-08209R3 

PLOS ONE

Dear Dr. Yan, 

I'm pleased to inform you that your manuscript has been deemed suitable for publication in PLOS ONE. Congratulations! Your manuscript is now being handed over to our production team.

Kind regards, 

on behalf of

Dr. Zsolt J. Balogh 

Academic Editor

PLOS ONE